# Anticholinesterase Inhibition, Drug-Likeness Assessment, and Molecular Docking Evaluation of Milk Protein-Derived Opioid Peptides for the Control of Alzheimer's Disease

Dawei Ji [1] , Jingying Ma [1] , Junyi Dai [2] , Min Xu [1] , Paul W. R. Harris [3] , Margaret A. Brimble [3] and Dominic Agyei [1,*]

1    Department of Food Science, University of Otago, Dunedin 9054, New Zealand; jidawei2008@gmail.com (D.J.); jingyingma6@gmail.com (J.M.); emmaemma181008@gmail.com (M.X.)
2    Department of Management, University of Otago, Dunedin 9054, New Zealand; daijunyi24@gmail.com
3    School of Biological Sciences, The University of Auckland, Auckland 1142, New Zealand; paul.harris@auckland.ac.nz (P.W.R.H.); m.brimble@auckland.ac.nz (M.A.B.)
*    Correspondence: dominic.agyei@otago.ac.nz; Tel.: +64-3-479-8735

**Abstract:** The drug-likeness and pharmacokinetic properties of 23 dairy-protein-derived opioid peptides were studied using SwissADME and ADMETlab in silico tools. All the opioid peptides had poor drug-like properties based on violations of Lipinski's rule-of-five. Moreover, prediction of their pharmacokinetic properties showed that the peptides had poor intestinal absorption and bioavailability. Following this, two well-known opioid peptides ($\beta_b$-casomorphin-5, $\beta_b$-casomorphin-7) from A1 bovine milk and caffeine (positive control) were selected for in silico molecular docking and in vitro inhibition study with two cholinesterase enzyme receptors important for the pathogenesis of Alzheimer's disease. Both peptides showed higher binding free energies and inhibitory activities to butyrylcholinesterase (BChE) than caffeine, but in vitro binding energy values were lower than those from the docking model. Moreover, the two casomorphins had lower inhibitory properties against acetylcholinesterase (AChE) than caffeine, although the docking model predicted the opposite. At 1 mg/mL concentrations, $\beta_b$-casomorphin-5 and $\beta_b$-casomorphin-7 showed promising results in inhibiting both cholinesterases (i.e., respectively 34% and 43% inhibition of AChE, and 67% and 81% inhibition of BChE). These dairy-derived opioid peptides have the potential to treat Alzheimer's disease via cholinesterase inhibition. However, appropriate derivatization may be required to improve their poor predicted intestinal absorption and bioavailability.

**Keywords:** casomorphins; opioid peptides; cholinesterase inhibitors; molecular docking; bioinformatics; Alzheimer's disease

## 1. Introduction

Alzheimer's disease, a chronic neurodegenerative disease, usually starts with mild memory loss and could culminate in impairment of cognitive ability, severe behavioral abnormalities, or death [1,2]. According to the cholinergic hypothesis, the cognitive decline in Alzheimer's disease is caused by the loss of the neurotransmitter acetylcholine. Therefore, inhibition of cholinestrases, the enzymes that degrade acetyl/butyrylcholine, is expected to increase the levels of systemic and circulating acetylcholine, thus increasing their availability to stimulate brain receptors for normal cognitive functions [3]. Consequently, one of the preferred treatment strategies to manage neurodegenerative conditions is the use of acetylcholinesterase (AChE) and butyrylcholinesterase (BChE) inhibitors [2,4]. The inhibition of these two key enzymes of Alzheimer's disease lessens the symptoms by increasing communication between cholinergic pathway activities and nerve endings [4]. Several synthetic drugs such as caproctamine, memantine, and tacrine, have been used to treat Alzheimer's disease, but these drugs present side effects such as dizziness, nausea,

vomiting, diarrhea, gastrointestinal disturbance, and hepatotoxicity [4]. Thus, it is important to discover new compounds from natural sources which could have fewer or no side effects and do not elicit physical dependence.

Some peptides found endogenously or produced via chemical and enzymatic synthesis have been shown to have opioid activity. Most of these opioid peptides can cross blood–brain barriers and have great potential for the development of drugs for pain alleviation and treatment and for controlling drug use and mood disorders, including anxiety and depression [5,6]. Moreover, the role of the opioid system in the pathogenicity of Alzheimer's disease and behavioral dysfunction, cognitive impairment, hyper-phosphorylation of tau, and beta-amyloid (A$\beta$) production has been reported [7]. This presupposes that active compounds that have inhibitory effects against the key enzyme players (e.g., AChE and BChE) in Alzheimer's disease could potentially control the onset and progression of Alzheimer's disease. In fact, fermented dairy products have been touted in some epidemiological reports as having the potential to protect against dementia and Alzheimer's disease, especially in the elderly [8–10].

It is well demonstrated that the opioid system may play a key role in the pathogenesis of dementia and Alzheimer's disease [7]. The first identified milk-derived peptides, $\beta_b$-casomorphin-7, mainly produced by the proteolysis of bovine A1 β-casein, have attracted increasing interest in recent years due to their potential neurological immune- or inflammation-related properties [11–13]. With removal of three, two, or one amino acid residue(s), $\beta_b$-casomorphin-4, -5, -6 are respectively derived from $\beta_b$-casomorphin-7 [14]. Among these, $\beta_b$-casomorphin-5 was reported to have the most potent opioid activity based on electrically stimulated guinea pig ileum (GPI) and mouse vas deferens (MVD) tests [14]. In contrast, milk-derived opioid peptides such as β-casomorphin-7 are cardioprotective in diabetic rats through an ability to induce hypoglycemic effects, lower oxidative stress, and regulate the levels of calcium ions [15]. In addition, certain food-derived opioid peptides have demonstrated an ability to elicit behavioral effects in rodents [16,17].

Therefore, research on the impact of the milk-derived peptides (e.g., $\beta_b$-casomorphin-4, -5, -6, and -7) on human health seems conflicting. However, to the best of our knowledge, the effects of exogenous milk-derived opioid peptides on Alzheimer's disease, mediated by cholinesterases inhibition, have not been studied intensively. Therefore, the current study aims to fill this gap by investigating the in silico and in vitro AChE and BChE inhibitory properties and drug-likeness of milk-derived opioid peptides, thereby laying the foundations for the development of exogenous peptide-based drugs for treating or improving the prognosis of Alzheimer's disease.

## 2. Materials and Methods

### 2.1. Preparation and Characterization of In Silico Opioid Peptides Used in This Study

The amino acid sequences of 23 opioid peptides from bovine and human milk were collected according to a recent literature review by Liu and Udenigwe [12] and converted into SMILES using the "SMILES" feature of BIOPEP-UWM (http://www.uwm.edu.pl/biochemia/index.php/pl/biopep, accessed on 21 July 2020) [18]. The SMILES of the opioid peptides having amino groups or methoxy groups at the C-terminal were obtained from PubChem (https://pubchem.ncbi.nlm.nih.gov/, accessed on 21 July 2020) or typed manually. The list of peptides used in this study is shown in Table 1.

**Table 1.** List of opioid peptides used in this study.

| Number | Opioid Peptide | Source | Sequence |
|---|---|---|---|
| 1 | $\beta_b$-casomorphin-4 | bovine milk β-casein | YPFP |
| 2 | $\beta_b$-casomorphin-5 | bovine milk β-casein | YPFPG |
| 3 | $\beta_b$-casomorphin-6 | bovine milk β-casein | YPFPGP |
| 4 | $\beta_b$-casomorphin-7 | bovine milk β-casein | YPFPGPI |
| 5 | $\beta_b$-casomorphin-8 (A1) | bovine milk A1 β-casein | YPFPGPIH |
| 6 | $\beta_b$-casomorphin-8 (A2) | bovine milk A2 β-casein | YPFPGPIP |
| 7 | neocasomorphin-6 | bovine milk β-casein | YPVEPF |
| 8 | $\alpha_b$-casein exorphin (1-7) | bovine milk α-casein | RYLGYLE |
| 9 | $\alpha_b$-casein exorphin (2-7) | bovine milk α-casein | YLGYLE |
| 10 | casoxin A | bovine milk κ-casein | YPSYGLN |
| 11 | casoxin B | bovine milk κ-casein | YPYY |
| 12 | casoxin C | bovine milk κ-casein | YIPIQYVLSR |
| 13 | α-lactorphin | bovine/human milk α-lactalbumin | YGLF-NH$_2$ |
| 14 | $\beta_b$-lactorphin | bovine milk β-lactoglobulin | YLLF-NH$_2$ |
| 15 | lactoferroxin A | bovine/human milk lactoferrin | YLGSGY-OCH$_3$ |
| 16 | lactoferroxin B | bovine/human milk lactoferrin | RYYGY-OCH$_3$ |
| 17 | lactoferroxin C | bovine/human milk lactoferrin | KYLGPQY-OCH$_3$ |
| 18 | $\beta_h$-casomorphin-4 | human milk β-casein | YPFV |
| 19 | $\beta_h$-casomorphin-5 | human milk β-casein | YPFVE |
| 20 | $\beta_h$-casomorphin-7 | human milk β-casein | YPFVEPI |
| 21 | $\beta_h$-casomorphin-8 | human milk β-casein | YPFVEPIP |
| 22 | casoxin D | human milk α-casein | YVPFPPF |
| 23 | $\alpha_h$-casomorphin | human milk α-casein | YVPFP |

### 2.2. In Silico Drug-Likeness Assessment of Opioid Peptides

In silico prediction of drug-likeness, as well as absorption, digestion, metabolism, excretion, and potential toxicity of the 23 opioid peptides, were computed using SwissADME (http://www.swissadme.ch, accessed on 22 July 2020) [19] and ADMETlab (http://admet.scbdd.com/, accessed on 22 July 2020) [20] platforms. SwissADME was used to predict physicochemical properties and drug-likeness (molecular weight, hydrogen bond acceptors/donors, lipophilicity, polarity, insolubility, insaturation, and flexibility) and data filtering based on Lipinski's rule-of-five [21]. A bioavailability radar was also obtained using the SwissADME platform. ADMETlab was used to compute pharmacokinetic indicators such as human intestinal absorption (HIA) behavior [22], potential metabolic behavior (substrate potential to permeability glycoprotein or multidrug resistance protein 1) [23], and potential to inhibit CYP3A4 (cytochrome P450 3A4).

### 2.3. Molecular Docking of Peptides

Molecular docking of the peptides was performed using Autodock CrankPep (ADCP), while Autodock vina was used for caffeine. The structures of $\beta_b$-casomorphin-5 and $\beta_b$-casomorphin-7 from bovine milk proteins were generated with Discovery Studio software version 2019 (Biovia Inc., San Diego, CA, USA). The structure of caffeine was downloaded from PubChem (https://pubchem.ncbi.nlm.nih.gov/, accessed on 21 July 2020). The receptors used in this study were downloaded from Worldwide Protein Data Bank (ID: 4bbz.pdb for BChE and ID: 2 × 8b.pdb for AChE). The water molecules and hetero atoms were removed, and polar hydrogen atoms were added prior to docking. For docking to AChE, a binding site was searched in a box of 20 × 20 × 20 Å with center on the coordinates x: 121, y: 109, and z: −134. For docking to BChE, a binding site was searched in a box of 20 × 20 × 20 Å with center on the coordinates x: −19, y: −40, and z: −25. All generated docking modes were evaluated according to affinity energy values. The in silico estimated dissociation constant (Ki) was calculated using affinity energy according to our previous study [24]. The DS 2019 software was utilized to view hydrogen bonds and hydrophilic, hydrophobic, and electrostatic interactions between residues at the AChE or BChE active sites and ligands.

### 2.4. Peptides Synthesis, Purification, and Characterization

All reagents were purchased as reagent grade and used without further purification. *N,N*-Diisopropylethylamine (DIPEA), dimethylformamide (DMF), piperidine, 1,2-ethanedithiol (EDT), trifluoroacetic acid (TFA), and triisopropylsilane (TIPS) were purchased from Sigma-Aldrich (St. Louis, MO, USA). *O*-(7-Azabenzotriazol-1-yl)-*N,N,N′,N′*-tetramethyluroniumhexafluorophosphate (HATU), Fmoc amino acids, and 2-chlorotrityl chloride resin were purchased from CS Bio (Shanghai, China). Trifluoroacetic acid was obtained from Oakwood Chemicals (Estill, SC, USA). Dichloromethane (DCM) was purchased from ECP chemicals (Auckland, New Zealand).

### 2.4.1. Synthesis of β-casomorphin 5

2-chlorotrityl chloride resin (180 mg, 0.2 mmol) was swollen in DCM (5 mL). A solution of Fmoc-Gly-OH (3 eq) and DIPEA (6 eq) in DCM (0.2 M concentration) was added to the resin and was left to shake for 16 h at room temperature. The peptide was elongated by manual solid-phase peptide synthesis (SPPS) using 20% piperidine in DMF (*v/v*, 2 × 5 min) for Fmoc removal and HATU (3eq)/DIPEA (3 eq) in DMF (1 × 30 min) for amino acid couplings. Upon completion, the resin was washed with DCM (10 × 5 mL) and dried under vacuum. The peptide was cleaved from resin on treatment with an acidic cocktail containing TFA/$H_2O$/EDT/TIPS (95:2:2:1, *v/v/v/v*) for 2 h at room temperature. The cleavage mixture was filtered and dried under a stream of $N_2$ gas. The crude material was precipitated with cold diethyl ether, and the product pellet was collected after centrifugation. The crude material was lyophilized to afford β-casomorphin 5 (100 mg, 86% yield, 95% purity, ESI-MS: *m/z* (M+H)$^+$ Calc. 580.7. Found 580.3), which was used without further purification.

### 2.4.2. Synthesis of β-casomorphin 7

2-chlorotrityl chloride resin (180 mg, 0.2 mmol) was swollen in DCM (5 mL). A solution of Fmoc-Ile-OH (3 eq) and DIPEA (6 eq) in DCM (0.2 M concentration) was added to the resin and was left to shake for 16 h at room temperature. The peptide was elongated and cleaved from the resin as described for β-casomorphin 5 to afford β-casomorphin 7 (100 mg, 63% yield, 90% purity, ESI-MS: *m/z* (M+H)$^+$ Calc. 790.9. Found 790.4) and used without further purification.

### 2.4.3. Peptide Characterization

Analytical HPLC was performed on a Waters Alliance using a Phenomenex (Sunnyvale, CA, USA) C8 Luna column (5 μm; 4.6 × 250 mm) at a flow rate of 1 mL/min using a linear gradient of 5% B to 95% B over 30 min, where solvent A was 0.1% TFA in $H_2O$ and B was 0.1% TFA in acetonitrile. Detection was at 210 and 254 nm. ESI MS direct infusion spectra were acquired with a Waters (Waltham, MA, USA) Quattro mass spectrometer operating in the positive mode using a solvent mixture of 1:1 $H_2O$:acetonitrile (*v/v*) containing 0.1% formic acid.

### 2.5. In Vitro Anticholinesterase Inhibitory Enzyme Activity of Casomorphins

In vitro anticholinesterase inhibition activities of β$_b$-casomorphin-5, β$_b$-casomorphin-7, and caffeine (positive control) were determined according to the methods developed by Ellman et al. [25]. Acetylcholinesterase (AChE, Sigma-Aldrich, USA) and butyrylcholinesterase (BChE, Sigma-Aldrich, USA) were employed for the inhibitory activity, using acetylthiocholineiodide (Sigma-Aldrich, USA) as substrate. In a 96-well plate was added 100 μL of substrate dissolved in 50 mM Tris-HCl (pH = 8.0) to give a final concentration ranging from 0 to 1000 μg/mL. To this, 25 μL of 15 mM acetylthiocholine iodide (dissolved in water) and 50 μL of 10 mM 5,5-dithio-bis-(2-nitrobenzoate) (DTNB, Sigma-Aldrich, USA) (dissolved in Tris-HCl, pH = 8.0) were added. After 5 min incubation in the dark at room temperature, 25 μL of 0.22 U/mL of AChE (dissolved in Tris-HCl, pH = 8.0) was added, and the absorbance was measured at 412 nm after 5 min of incubation

in the dark at room temperature. Enzyme activity and inhibition were calculated based on Equations (1) and (2).

$$\%enzyme\ activity = \frac{V}{V_{max}} \times 100\% = \frac{OD_{drug} - OD_{Blank}}{OD_{control} - OD_{Blank}} \times 100\% \tag{1}$$

where $V$ = reaction velocity, $V_{max}$ = maximum reaction velocity, $OD$ = optical density at 412 nm.

$$\%enzyme\ inhibition = 1 - enzyme\ activity\% \tag{2}$$

The experimental dissociation constants ($K_i$) were calculated using the $IC_{50}$-to $K_i$ converter (https://bioinfo-abcc.ncifcrf.gov/IC50_Ki_Converter/index.php, accessed on 22 July 2020) developed by Cer et al. [26]. The half-maximal inhibitory concentration ($IC_{50}$) was estimated using the 'Dose–Response–Inhibition' feature in GraphPad Prism 8.

## 3. Results and Discussion

### 3.1. Hypothesis and Justification

This study hypothesized that in silico prediction and molecular docking tools could be used to forecast the physicochemical properties and medicinal chemistry friendliness of milk-protein-derived opioid peptides with anticholinesterase properties. Indeed, Mondal et al. [27] recently used computational techniques to design and develop an octapeptide (NFDVLTEQ) acetylcholinesterase inhibitor that is non-toxic, has good serum stability, can cross the blood–brain barrier, and have inhibiting effects on amyloid aggregation. Moreover, Trivedi et al. [28] demonstrated using human gastrointestinal epithelial cells in vivo that opioid peptides derived from foods such as milk caseins and wheat gliadin can interact with opioid receptors to elicit a response. It is also known that some opioid peptides activate enteric opioid receptors in the gastrointestinal system [29]. Together, these studies show the possibility of using computational techniques to predict the potential of milk-derived opioid peptides to control Alzheimer's disease by inhibiting the activity of acetylcholinesterases.

### 3.2. Drug-Likeness and Related Properties of Opioid Peptides

The drug-likeness of 23 opioid peptides from bovine and human milk were predicted in silico using 'Druglikeness analysis' and 'ADME/T evaluation' of ADMETlab [20]. Milk proteins are an excellent and abundant source of peptides [30], and most of them can be studied for their drug-likeness and cholinesterase inhibiting properties. The fact that these peptides are derived from food—a natural and safe source—is considered an important feature in developing alternative drugs for controlling stress-related health conditions and cognitive impairments [31].

#### 3.2.1. In Silico Evaluation of Physicochemical Properties of the Opioid Peptides

As shown in Table 2, the molecular weight of the 23 opioid peptides ranged from 498 to 1251. The molecular weight (498) of only one opioid peptide (α-lactorphin) was less than 500 following the rule-of-five [21], while most (22) opioid peptides had a molecular weight greater than 500. The high molecular weight is related to poorer intestinal and blood–brain barrier permeability [21], indicating that the opioid peptides are likely to be orally active. On the other hand, most (18) and some (9) of the opioid peptides had excessive numbers of hydrogen bond donor groups (>5) and hydrogen bond acceptor groups (>10), and this further impairs their permeability across membrane bilayers [21].

**Table 2.** In silico drug-likeness, absorption, distribution, metabolism, and excretion profile of the opioid peptides from bovine and human milk.

| No. | | Rule-of-5 | | | | | HIA (+/−) | Pgp-Substrate (Y/N) | BBB (+/−) | CYP3A4-Inhibitor (Y/N) |
|---|---|---|---|---|---|---|---|---|---|---|
| | | Molecular weight | HBD | HBA | LogP | Solubility (µg/mL) | | | | |
| | Optimal range | ≤500 | ≤5 | ≤10 | ≤5 | >10 | | | | |
| 1 | $\beta_b$-casomorphin-4 | 523 | 4 | 6 | 1.056 | 158 | (−) | N | (−) | N |
| 2 | $\beta_b$-casomorphin-5 | 580 | 5 | 7 | 0.172 | 341 | (−) | Y | (−) | Y |
| 3 | $\beta_b$-casomorphin-6 | 677 | 5 | 8 | 0.163 | 300 | (−) | Y | (−) | Y |
| 4 | $\beta_b$-casomorphin-7 | 790 | 6 | 9 | 0.694 | 217 | (−) | Y | (−) | Y |
| 5 | $\beta_b$-casomorphin-8 (A1) | 927 | 8 | 11 | 0.145 | 272 | (−) | Y | (−) | Y |
| 6 | $\beta_b$-casomorphin-8 (A2) | 887 | 6 | 10 | 0.685 | 216 | (−) | Y | (−) | Y |
| 7 | neocasomorphin-6 | 751 | 7 | 9 | 0.547 | 245 | (−) | Y | (−) | Y |
| 8 | $\alpha_b$-casein exorphin (1-7) | 913 | 14 | 12 | −0.945 | 438 | (−) | Y | (+) | Y |
| 9 | $\alpha_b$-casein exorphin (2-7) | 757 | 10 | 10 | 0.307 | 374 | (−) | Y | (+) | N |
| 10 | casoxin A | 813 | 11 | 12 | −2.745 | 415 | (−) | Y | (−) | Y |
| 11 | casoxin B | 605 | 7 | 8 | 1.204 | 159 | (−) | N | (−) | N |
| 12 | casoxin C | 1251 | 17 | 16 | −1.52 | 331 | (−) | Y | (−) | Y |
| 13 | $\alpha$-lactorphin | 498 | 6 | 6 | 0.122 | 255 | (−) | Y | (+) | Y |
| 14 | $\beta_b$-lactorphin | 554 | 6 | 6 | 1.537 | 123 | (+) | Y | (+) | N |
| 15 | lactoferroxin A | 673 | 9 | 11 | −1.891 | 430 | (−) | Y | (−) | N |
| 16 | lactoferroxin B | 735 | 10 | 11 | −1.044 | 257 | (−) | Y | (+) | Y |
| 17 | lactoferroxin C | 882 | 10 | 13 | −1.13 | 380 | (−) | Y | (−) | Y |
| 18 | $\beta_h$-casomorphin-4 | 525 | 5 | 6 | 1.206 | 162 | (−) | Y | (−) | Y |
| 19 | $\beta_h$-casomorphin-5 | 654 | 7 | 8 | 0.556 | 237 | (−) | Y | (−) | Y |
| 20 | $\beta_h$-casomorphin-7 | 864 | 8 | 10 | 1.077 | 210 | (−) | Y | (−) | Y |
| 21 | $\beta_h$-casomorphin-8 | 961 | 8 | 11 | 1.068 | 207 | (−) | Y | (−) | Y |
| 22 | casoxin D | 866 | 6 | 19 | 1.915 | 125 | (−) | Y | (−) | Y |
| 23 | $\alpha_h$-casomorphin | 622 | 5 | 7 | 1.197 | 199 | (−) | Y | (−) | Y |
| 24 | Caffeine | 194 | 6 | 0 | −1.029 | 11,435 | (+) | N | (+) | N |

Abbreviations: HBD, hydrogen bond donors; HBA, hydrogen bond acceptors; Log P, the logarithm of compound partition coefficient between *n*-octanol and water; HIA, human intestinal absorption; Pgp, P-glycoprotein; BBB, blood–brain barrier; CYP3A4, cytochrome P450 3A4.

In contrast, the lipophilicity of all (23) opioid peptides, calculated by the partition coefficient between n-octanol and water [19], was less than 5 (range of −2.745 to 1.915) following the rule-of-five, indicating the concentration ratio of the opioid peptides between the two liquid phases. Poor solubility is often considered a major issue in drug discovery and development and occurs in 75% of drug candidates [32]. Lipinski [33] described a qualitative estimation of the solubility class according to the following scale: low solubility < −10 µg/mL < moderate solubility < −60 µg/mL < high solubility. As shown in Table 2, the solubility of all (23) opioid peptides was high, ranging from 123 to 438 µg/mL.

3.2.2. In Silico Evaluation of the Pharmacokinetic Properties of the Opioid Peptides

The parameter, human intestinal absorption (HIA), is estimated by random forest algorithms, and a value of 30% of HIA value (%) is used as the criterion to divide new drug entities into two classes: poor absorption (HIA (−)) and good absorption (HIA (+)) [34]. It can be seen in Table 2 that, except for β-lactorphin from bovine milk, the remaining 22 opioid peptides showed poor absorption. The predicted poor absorption and transport properties for most of the opioid peptides in this study reflect the findings of some in vivo experimental studies. For example, the intestines of human adults are less permeable to the absorption of casomorphins, and casomorphins have not been detected in adult human postprandial blood [9].

Analysis using the SwissADME platform shed further light on the oral bioavailability of these opioid peptides. The bioavailability score predicts the probability of a compound

having at least 10% oral bioavailability in rats or measurable Caco-2 permeability. The bioavailability score is calculated using topological polar surface area (TPSA), total charge, and violations of the Lipinski rules to estimate a compound's probability of being orally bioavailable. The bioavailability score puts the compounds into four classes with probabilities of 11%, 17%, 56%, or 85%. A score of 17% or below is a fail. A 55% or above score means the compound can be orally bioavailable [19]. In this study, the bioavailability score of most of the opioid peptides was less than or equal to 0.17 (17%), and only three opioid peptides (#1, bovine β-casomorphin-4; #13, human/bovine α-lactorphin; #18, human β-casomorphin-4) have moderate bioavailability (0.55). The bioavailability radar plots generated from SwissADME (see Figure 1) show that the lipophilicity, solubility, and saturation of most of the 23 opioid peptides are optimal. However, other parameters such as flexibility, polarity, and molecular weight make these peptides unsuitable for oral administration.

In the development of drugs that must function in the central nervous system (CNS), the capability of the drug candidate to cross the blood–brain barrier (BBB) is one of the most important properties [35]. This study estimated BBB using support vector machine (SVM) algorithms and divided it into two classes: BBB+ and BBB− [20]. As shown in Table 2, five opioid peptides (i.e., $\alpha_b$-casein exorphins 1-7 and 2-7, α-lactorphin, lactoferroxin B, and $\beta_b$-lactorphin) were predicted to be able to cross the blood–brain barrier (BBB+), while the other 19 opioid peptides had a negative blood–brain barrier crossing potential (BBB−).

P-glycoprotein (Pgp), a cell membrane protein, can shunt drug compounds from many cells and is associated with the drug's absorption, excretion, drug–drug interactions, and CNS effects [23]. Pgp protects cells against potentially toxic compounds through increasing renal and biliary elimination, but it could also limit the cytosolic accumulation of drug molecules to decrease their intestinal absorption and bioavailability [36]. In this study, most (21) of the opioid peptides from bovine and human milk were Pgp-substrates, implying that these opioid peptides were likely to have poor intestinal absorption and bioavailability (Table 2).

The cytochrome P450 (CYP) is a ubiquitous enzyme family that strongly influences drug discovery and development [37,38]. One of the most important human CYP isoforms, CYP3A4, is a critical enzyme for drug metabolism, improving the protection of tissues and organisms through synergistically eliminating drug molecules with P-glycoprotein [39]. As shown in Table 2, most (18) opioid peptides were predicted to be inhibitors of CYP3A4. Due to the accumulation of the drug or its metabolites with poorer clearance, the inhibition of opioid peptides could lead to pharmacokinetics-related drug–drug interactions, which might cause toxicity or other unwanted adverse effects.

### 3.3. Molecular Docking of Peptides to Substrates

Following in silico predictions, two casomorphins ($\beta_b$-casomorphin-5 and $\beta_b$-casomorphin-7) were chosen for further studies. These two peptides were chosen because their opioid properties have been established, and they are also two peptides implicated in the a1/a2 debate. In addition, they are widely available commercially for use in in vitro studies. First, the inhibitory effect of these two casomorphins (β-casomorphin-5 and-7) and caffeine (positive control) on AChE and BChE were further studied using molecular docking to investigate their interactions at the molecular level. Table 3 shows the inhibition constant ($K_i$) of casomorphins and caffeine to AChE and BChE obtained from the docking calculations. The estimated $K_i$ values suggested that $\beta_b$-casomorphin-5 and $\beta_b$-casomorphin-7 are expected to have a higher inhibitory effect in BChE and AChE than caffeine.

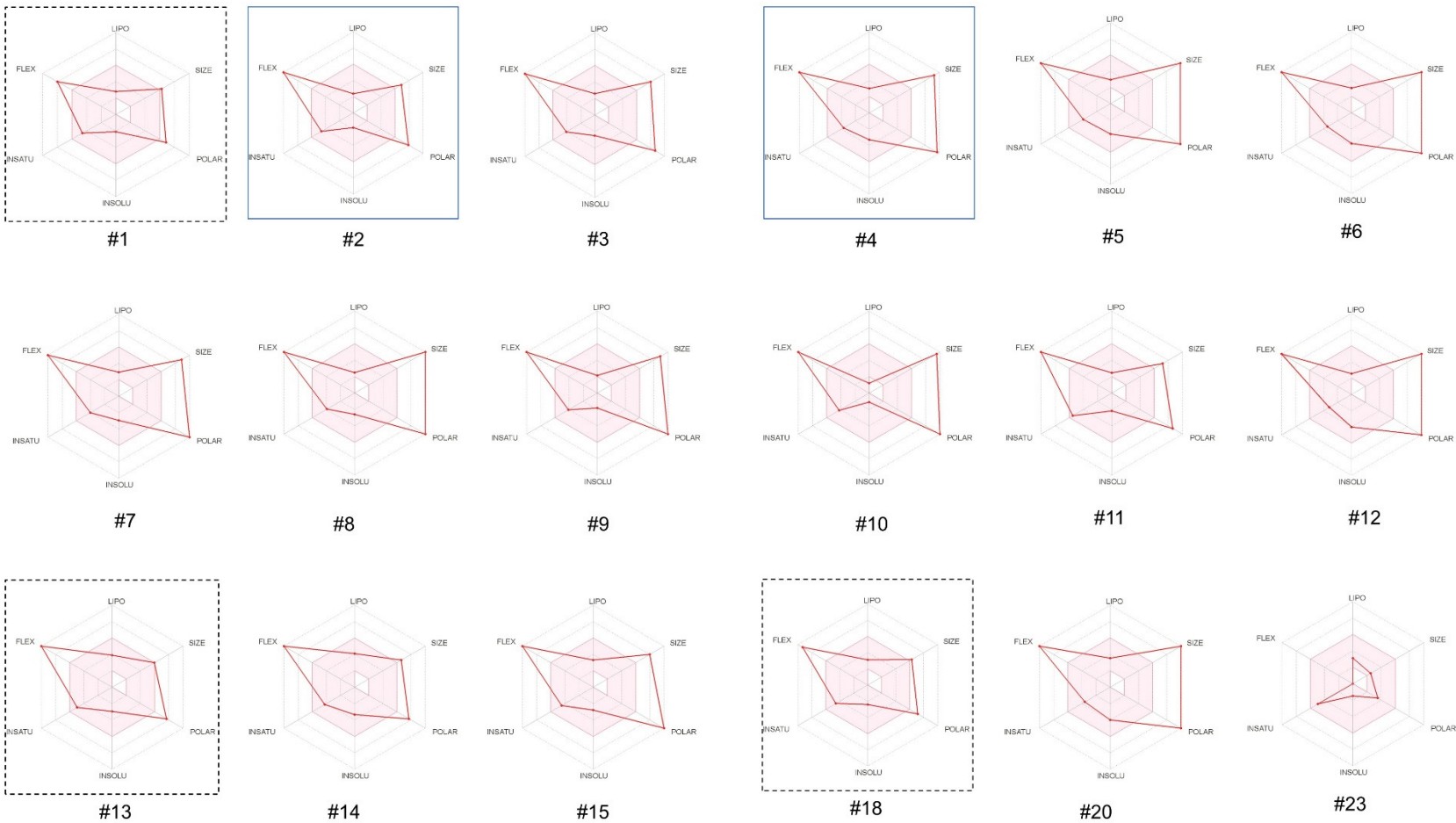

**Figure 1.** Representative bioavailability radar plots of the opioid peptides and caffeine (compound #23). The colored zone is the suitable physicochemical space for oral bioavailability. LIPO, lipophilicity; SIZE, molecular weight; POLAR, polarity; INSOLU, insolubility; INSATU, insaturation; FLEX, flexibility. Compounds in dashed squares had moderate bioavailability scores (0.55). Compounds #2 and #4 are β-casomorphin-5 and β-casomorphin-7, respectively. Refer to Table 2 for the chemical names and identities corresponding to the other compounds in this figure.

**Table 3.** Binding free energy of compounds to AChE (PDB: 2 × 8B) and BChE (PDB: 4BBZ) calculated by molecular docking, as well as computational dissociation constant.

| Ligand | AChE | | BChE | |
|---|---|---|---|---|
| | Binding free energy (kcal/mol) | Estimated $K_i$ (μM) | Binding free energy (kJ/mol) | Estimated $K_i$ (μM) |
| $\beta_b$-casomorphin-5 | −14.6 | $1.93 \times 10^{-5}$ | −14.8 | $1.38 \times 10^{-5}$ |
| $\beta_b$-casomorphin-7 | −10.1 | 0.04 | −14.4 | $2.71 \times 10^{-5}$ |
| Caffeine | −6.7 | 12.12 | −6.5 | 16.99 |

In AChE, the best poses of casomorphins and caffeine were mainly stabilized by hydrophobic interactions and hydrogen bonds (Figure 2). The structure of human acetylcholinesterase consists of four subunits, including the esteratic site, anionic subsite, oxyanion hole, and the acyl pocket [40]. The esteratic site is a narrow, long, hydrophobic gorge (approximately 20 Å deep) containing the catalytic triad (Ser203, His447, and Glu334) that hydrolyzes acetylcholine (ACh) to acetate and choline [40,41]. As shown in Table 4, all four compounds interact with the catalytic site through His447. The peripheral anionic site is located at the rim of the narrow gorge comprising a set of aromatic residues: Tyr72, Asp74, TRP86, Tyr124, Ser 125, Trp286, Tyr337, Phe338, and Tyr341 [42]. Of these aromatic residues, TRP86 was important for the enzyme activity because the acetylcholinesterase activity decreased 3000-fold when TRP86 was replaced by alanine [40]. Figure 2 and Table 4 show that $\beta_b$-casomorphin-5 and $\beta_b$-casomorphin-7 interact with 4 and 6 residues of this area, respectively, which is more than caffeine (2 residues), but all three compounds interact with the most important residues, TRP86. The oxyanion hole, consisting of Gly121, Gly122, and Ala204, can help stabilize the tetrahedral intermediate form of the substrate (Ach) during catalysis through contributing hydrogen bond donors [40]. Moreover, as seen in Figure 2 and Table 4, all four compounds interact with the oxyanion hole. The acyl pocket, containing Phe295 and Phe297 as gatekeepers, limits the dimension of ligands entering the active site [40]. $\beta_b$-casomorphin-5 and $\beta_b$-casomorphin-7 interacted with the acyl pocket, while caffeine had no interaction with this area (Figure 2 and Table 4).

**Table 4.** AChE (PDB: 2 × 8B) interact with compounds (□: hydrophobic interactions; △: hydrogen bonds; •: electrostatic bonds; ∗: unfavorable bonds).

| | $\beta_b$-casomorphin-5 | $\beta_b$-casomorphin-7 | Caffeine |
|---|---|---|---|
| PRO31 | | □ | |
| TYR72 | | ∗ | |
| ASP74 | △ | • | |
| TRP86 | □ | □ | □ |
| GLY120 | | △ | |
| GLY121 | △ | △ | △ |
| GLY122 | | △ ∗ | |
| TYR124 | | △ | |
| GLY126 | △ | | |
| GLU202 | △ • | △ | △ |
| TRP236 | | □ | |
| VAL294 | | □ | |
| PHE295 | □ | △ | |
| PHE297 | □ | | |
| TYR337 | | □ | □ |
| PHE338 | □ | □ | |
| TYR341 | □ | □ | |
| TRP439 | △ | | |
| HIS447 | □ | □ | □ |
| GLY448 | | | △ |
| TYR449 | △ | | |

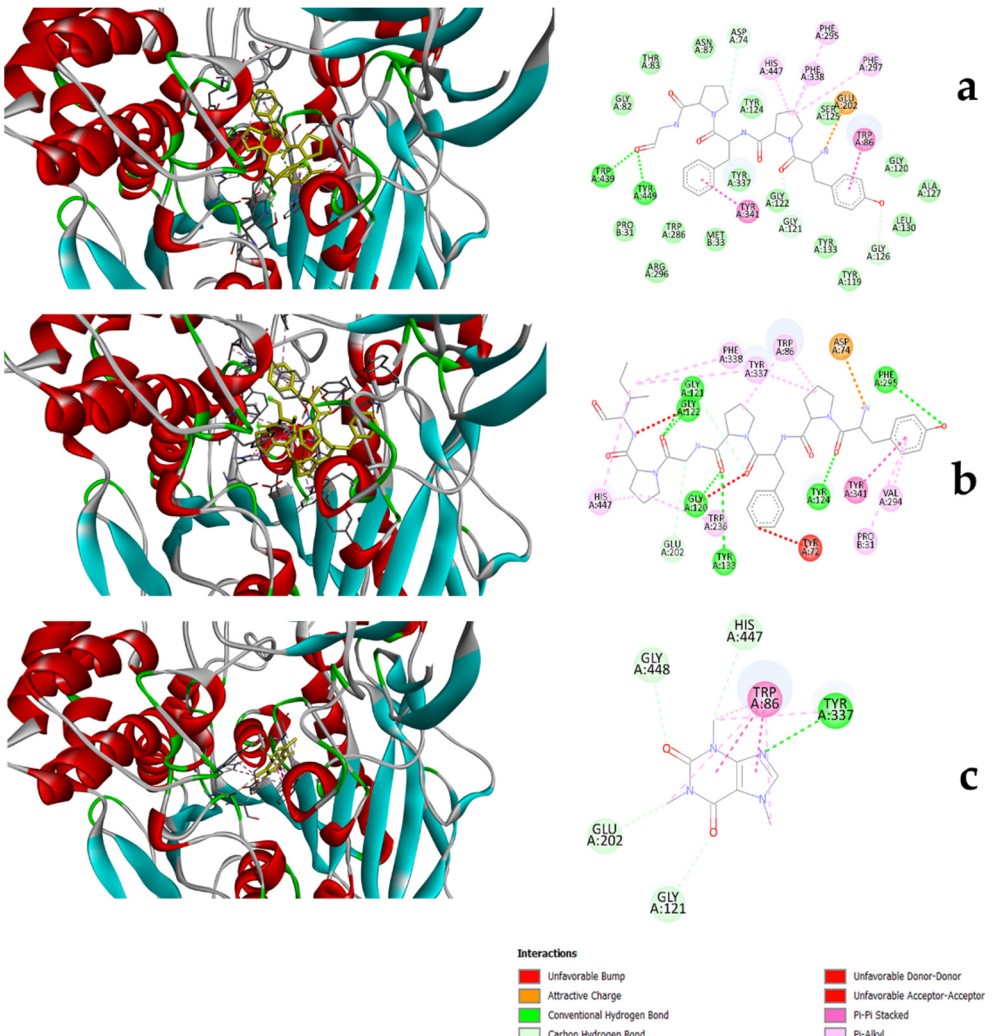

**Figure 2.** Predicted binding mode of opioid peptides to AChE (PDB: 2 × 8B). (**a**) β<sub>b</sub>-casomorphin-5, (**b**) β<sub>b</sub>-casomorphin-7, and (**c**) caffeine. The best-scored docking pose is shown.

The crystal structure of human butyrylcholinesterase is also well characterized through X-ray studies and consists of a catalytic triad (Glu325, Ser198, and His438), an anion site (Trp82, Tyr128, and Phe329), a peripheral anionic site (Asp70 and Tyr332), an oxyanion hole (Gly116, Gly117, and Ala199), and an acyl pocket (Val288, and Leu286) [43]. As shown in Table 5 and Figure 3, β<sub>b</sub>-casomorphin-5, β<sub>b</sub>-casomorphin-7, and caffeine have interactions with the catalytic triad (His438, Ser198) and the peripheral anionic site (Asp70 and Tyr332). Moreover, the opioid peptides showed interactions with the anion site (Trp82 and Phe329) but not with the oxyanion hole. Notably, only β<sub>b</sub>-casomorphin-5 interacts with the acyl pocket (Leu286).

**Table 5.** BChE (PDB: 4BBZ) interaction with compounds (□: hydrophobic interactions; △: hydrogen bonds; ●: electrostatic bonds).

| | β<sub>b</sub>-casomorphin-5 | β<sub>b</sub>-casomorphin-7 | Caffeine |
|---|---|---|---|
| GLN67 | △ | | |
| ASN68 | △ | | |
| ILE69 | △ | | |
| ASP70 | △ | | △ |

**Table 5.** *Cont.*

|  | β_b-casomorphin-5 | β_b-casomorphin-7 | Caffeine |
|---|---|---|---|
| TRP82 | □ | □ ● | □ |
| PRO84 | △ |  |  |
| THR120 | □ |  |  |
| GLU197 |  | △ ● |  |
| SER198 | △ |  |  |
| THR284 |  | △ |  |
| PRO285 |  | △ |  |
| LEU286 | □ |  |  |
| ALA328 |  | □ | □ |
| PHE329 | □ |  |  |
| TYR332 | □ | △ □ | △ |
| TRP430 |  |  |  |
| MET437 |  |  | □ |
| HIS438 | △ | △ ● | □ ● |
| TYR440 |  |  | □ |

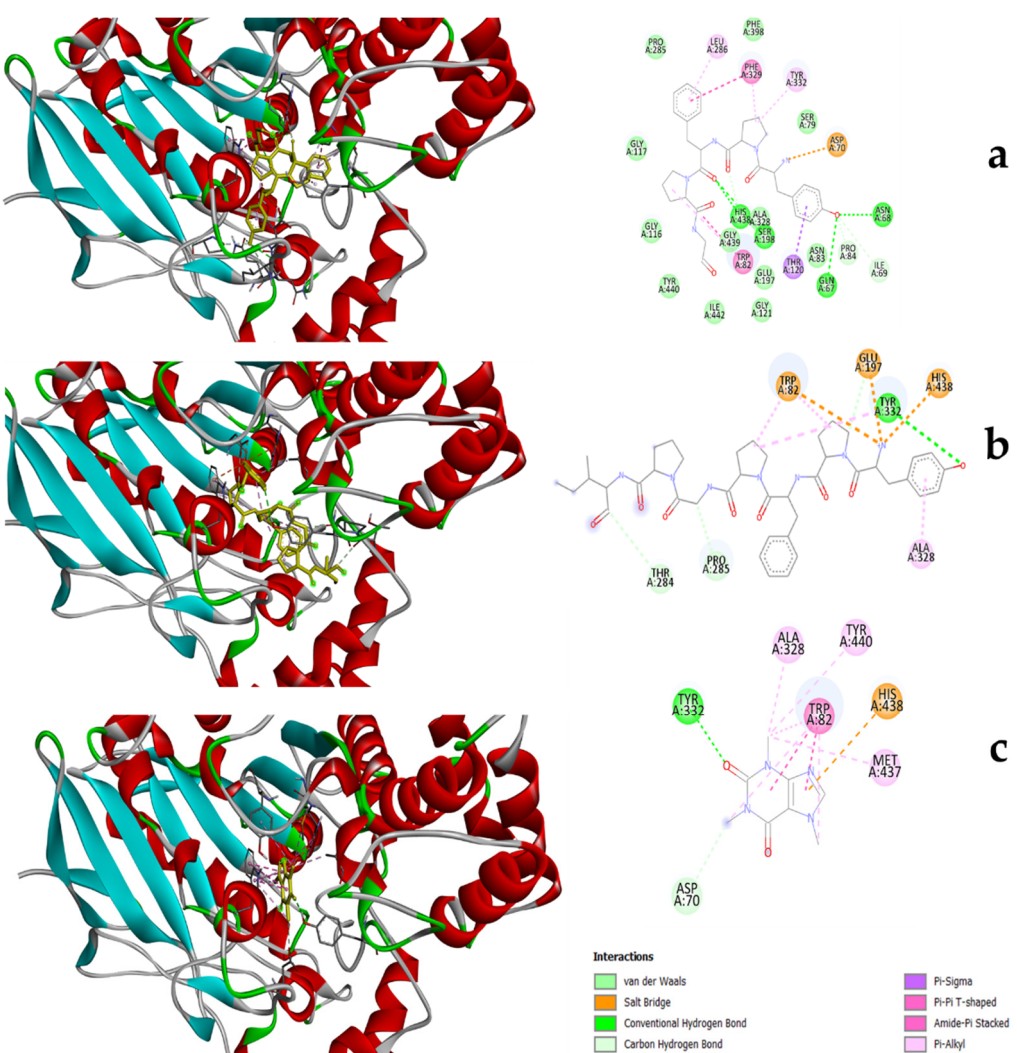

**Figure 3.** Predicted binding mode of opioid peptides to BChE (PDB: 4BBZ). (**a**) β_b-casomorphin-5; (**b**) β_b-casomorphin-7; and (**c**) caffeine. The best-scored docking pose is shown.

### 3.4. In Vitro Cholinesterase Inhibition by β-casomorphin-5 and β-casomorphin-7

The casomorphins reported in this study had cholinesterase inhibition behavior, which suggests that they may play a role in controlling Alzheimer's disease. This finding is in line

with other studies that have reported cholinesterase inhibition behavior in dairy products (e.g., cholinesterase inhibition by hydrolyzed ricotta in chocolate dessert [44]).

As seen in Table 6, β-casomorphin-5 and β-casomorphin-7 showed better activity as BChE inhibitors, with $IC_{50}$ of 238.5 μg/mL (411.2 μm) and 194.7 μg/mL (246.5 μm), respectively, whereas caffeine only had an $IC_{50}$ of 949.7 μg/mL (4895.4 μM). However, both peptides had lower AChE inhibiting activities, with $IC_{50}$ of 7151 μg/mL (12329.3 μm) and 4255 μg/mL (5386.1 μm), respectively, compared with caffeine which inhibited AChE at an $IC_{50}$ of 7.4 μg/mL (38.1 μm). The % inhibition of AChE by β-casomorphin-5 and β-casomorphin-7 at 1 mg/mL (Table 6) were relatively higher than values reported for enzymatically generated AChE inhibiting peptides from mussel by-products (≤29.6% from 1 mg/mL substrate [45]). Moreover, the findings show AChE inhibition that is comparable to those reported in anchovy protein hydrolysates (10–60% inhibition, at concentrations higher than those used in this study (i.e., 100–400 mg/mL, [46]), or fish protein hydrolysates (10.5 –40.5% inhibition at 20–50 mg/mL [47]).

**Table 6.** Experimental dissociation constants, inhibition (%), IC50 values, and equivalents of the opioid peptides against AChE and BChE.

| Ligand | Experimental $K_i$ (μM) | | Inhibition (%) at 1000 μg/mL | | $IC_{50}$ (μg/mL) | |
|---|---|---|---|---|---|---|
| | AChE | BChE | AChE | BChE | AChE | BChE |
| β-casomorphin-5 | 3814.6 | 127.2 | 34% | 67% | 7151 | 238.5 |
| β-casomorphin-7 | 1666.4 | 76.3 | 43% | 81% | 4255 | 194.7 |
| Caffeine | 11.8 | 1514.0 | 97% | 48% | 7.4 | 949.7 |

Although several studies have reported the inhibition of BChE by various compounds (e.g., see [48]), studies reporting inhibition of BChE by milk-derived peptides are scant. In our study, β-casomorphin-5 and β-casomorphin-7 at 1 mg/mL inhibited BChE significantly (i.e., 67% and 81%, respectively).

When data from Tables 3 and 6 are compared, it is observed that the estimated $K_i$ of caffeine (12.12 μm) was similar to the experimental result (11.8 μm). However, the results given by Autodock CrankPep (ADCP) suggested that $β_b$-casomorphin-5 and $β_b$-casomorphin-7 are expected to have much higher inhibitory behavior on AChE than caffeine does. However, the available experimental data (see Table 6) showed the reverse and that the inhibitory effects of the casomorphins were rather lower than that of caffeine. In the in vitro experiments, the estimated $K_i$ values of casomorphins and caffeine were much higher than values from in silico predictions. This finding demonstrates a failure of the docking model to reflect wet experimental results.

In silico molecular docking and bioinformatics approaches used to predict the bioactivity of peptides always require experimental validation [49]. However, binding energies obtained from computational chemistry do not always corroborate in vitro findings. For example, despite in silico and molecular docking findings showing the antimicrobial properties of rapeseed seed storage proteins cruciferin and napin, only napin demonstrated antimicrobial properties in vitro [50]. Moreover, Ramírez and Caballero [51] reported that molecular docking to predict the binding energies of enantiomers is unreliable. Such research outcomes can be attributed to the limitation of in silico tools [49] or conditions of materials (especially enzymes and reagents) used in the in vitro experiment [50]. Notwithstanding, the works of Mondal et al. [27] demonstrated the suitability of docking models in screening a wide range of compounds for their medicinal properties and potential drug-likeness. Moreover, quantitative structure–activity relationship (QSAR) models and docking studies are important in drug discovery [52] and for identifying the pharmacological properties [53], and structural properties important in both substrates and ligands for enhanced biological activities [54,55].

The reaction velocity profiles and kinetic constants ($K_m$ and $V_{max}$) for acetylcholinesterase and butyrylcholinesterase with acetylthiocholine iodide as substrate are shown in Figure 4. These findings suggest that acetylthiocholine iodide is a better substrate for AChE than BChE, considering the lower $K_m$ value of 0.84 mM obtained for the former.

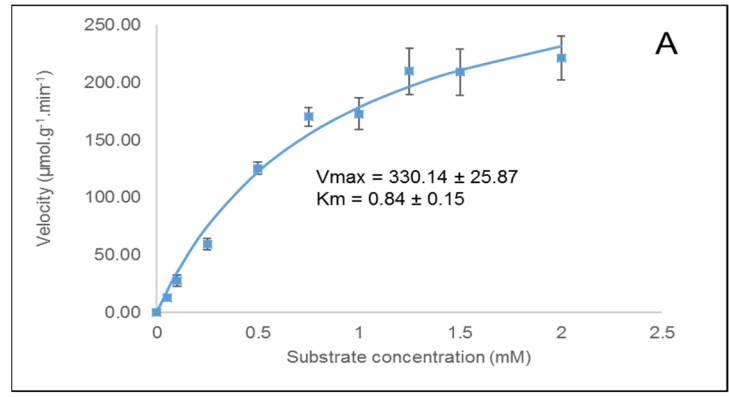

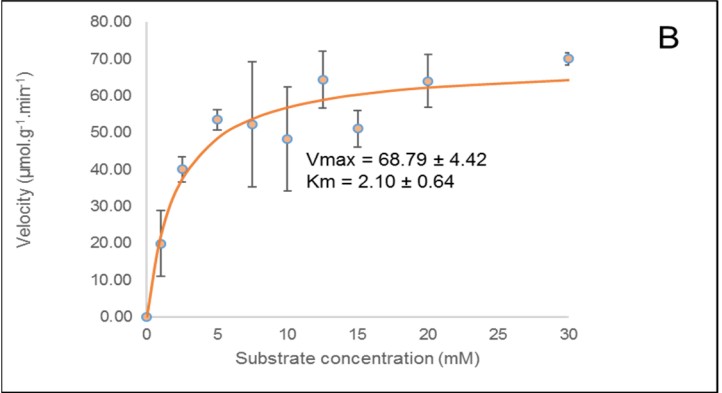

**Figure 4.** Kinetic constants $K_m$ and $V_{max}$ for acetylcholinesterase (**A**) and butyrylcholinesterase (**B**) with acetylthiocholine iodide as substrate.

### 4. Conclusions

In this study, the potential to use milk-derived opioid peptides as drug leads for treating Alzheimer's disease through the mechanisms of cholinesterase inhibition was investigated in silico and in vitro. Most of the 23 opioid peptides did not have desirable drug-like properties, as determined by their violations of one or more of Lipinski's rule-of-five. Furthermore, most peptides had poor intestinal absorption and bioavailability, as predicted by human intestinal absorption, blood–brain barrier transport, and potential to be a substrate for P-glycoprotein or inhibit CYP3A4.

Molecular docking of two well-known opioid peptides ($\beta_b$-casomorphin-5 and $\beta_b$-casomorphin-7) and caffeine (positive control) to acetylcholinesterase (AChE) and butyrylcholinesterase (BChE), two enzyme receptors for Alzheimer's disease showed both casomorphins having relatively high binding energies to the receptors (estimated $K_i$ (mM) of $1.93 \times 10^{-5}$ and $1.38 \times 10^{-5}$ for $\beta_b$-casomorphin-5, and 0.04 and $2.71 \times 10^{-5}$ for $\beta_b$-casomorphin-7 interactions with AChE and BChE, respectively). Binding in both cases of peptides and receptors was mostly via hydrogen bonding and hydrophobic interactions. However, in vitro experiments showed significantly high $K_i$ values for AChE (3814.6 and 1666.4 mm for $\beta_b$-casomorphin-5 and $\beta_b$-casomorphin-7, respectively) and BChE (127.2 and 76.3 mm for $\beta_b$-casomorphin-5 and $\beta_b$-casomorphin-7, respectively) and these differed significantly from the in silico findings. In follow-up studies, it will be interesting to study other docking methods or platforms.

At 1 mg/mL concentrations, $\beta_b$-casomorphin-5 and $\beta_b$-casomorphin-7 showed promising results in inhibiting both cholinesterases (i.e., respectively 34% and 43% inhibition of

AChE, and 67% and 81% inhibition of BChE). The inhibiting properties of $\beta_b$-casomorphin-5 and $\beta_b$-casomorphin-7 to both cholinesterases provide encouraging data on their potential in treating Alzheimer's disease. However, their poor intestinal absorption and bioavailability predicted in silico need to be overcome via derivatization with appropriate functional groups.

**Author Contributions:** Conceptualization, D.A. and D.J.; methodology, D.A. and D.J.; software, D.J and J.D.; validation, D.J.; formal Analysis, D.J. and D.A.; investigation, D.J.; resources, P.H and M.A.B.; writing—original draft preparation, D.J., D.A., J.M., M.X., P.W.R.H. and M.A.B.; writing—review and editing, D.A. and J.M.; visualization, J.M.; supervision, D.A. All authors have read and agreed to the published version of the manuscript.

**Funding:** This research received no external funding.

**Institutional Review Board Statement:** Not applicable.

**Informed Consent Statement:** Not applicable.

**Data Availability Statement:** The data presented in this study are available on request from the corresponding author.

**Acknowledgments:** The authors acknowledge the support of Aladin Bekhit in the preparation of this paper.

**Conflicts of Interest:** The authors declare no conflict of interest.

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
