# Peer review of "Anticholinesterase Inhibition, Drug-Likeness Assessment, and Molecular Docking Evaluation of Milk Protein-Derived Opioid Peptides for the Control of Alzheimer’s Disease"

_2624-862X, doi:10.3390/dairy3030032_

Round 1
Reviewer 1 Report
One specific group of bioactive peptides, known as β-casomorphins, are opioid-agonists for μ-receptors and have been suggested to assume an active role in the development of various non-communicable diseases, including diabetes mellitus, cardiovascular diseases, neurological disorders, pulmonary inflammation, to name a few.
The data mined from in vitro and ex vivo trials on the health impact of β-casomorphins is both inconclusive and limited to completely support the possible adverse or potential beneficial health effects of β-casomorphins.However, since the overall concern with β-casomorphins appears debatable and not well defined, more experimental trials are required to investigate the metabolic pathways of these identified peptides, their release during digestion, and subsequent fate after the digestion process. Consequently, repeatability of the findings under a number of other laboratory conditions is required before the data can be fully substantiated. Due to the rapidly evolving nature of the issue and emerging studies in this field, further exploration into the bioactivity of β-casomorphins is warranted.
The results of the study of this manuscript prove that at 1 mg/mL concentrations, βb-casomorphin-5 and βb-casomorphin-7 in inhibiting both cholinesterases . The results provide encouraging data on their potential in treating Alzheimer's disease.
The results are properly described and confronted with the research of other authors. The aim of the research established by the authors was properly edited and confirmed by research. The results are properly described and confronted with the research of other authors. The aim of the research established by the authors was properly edited and confirmed by research. Figures and tables are legible and correctly written.
The manuscript applies to milk-derived peptides (e.g., βb-casomorphin -4, -5, -6, and -7), i.e. it is part of the scope and purpose of Dairy.
Author Response
Authors’ response: We thank the Reviewer for the positive comments on our manuscript.

Reviewer 2 Report
The objective of this studied was investigated the pharmacokinetic properties and drug-likeness of 23 dairy protein derived opioid peptides using SwissADME and ADMETlab in silico tool with two cholinesterase enzyme receptors important for the pathogenesis of Alzheimer's disease in vitro.
Between lines 223-265, the manuscript has problems with interline.
The table 2 must be improved.
This information of table 2 "Abbreviations: HBD, hydrogen bond donors; HBA, hydrogen bond acceptors; Log P, the logarithm of compound partition coefficient between n-octanol and water; HIA, human intestinal absorption; Pgp, P-glycoprotein; BBB, Blood-Brain Barrier; CYP3A4, Cytochrome P450 3A4" in foot of the table.
The line 217 erase Lipinski.
The conclusion must be improved. Erase line 481-483. Most of…
Author Response
Reviewer 2
The objective of this studied was investigated the pharmacokinetic properties and drug-likeness of 23 dairy protein derived opioid peptides using SwissADME and ADMETlab in silico tool with two cholinesterase enzyme receptors important for the pathogenesis of Alzheimer's disease in vitro.
Between lines 223-265, the manuscript has problems with interline.
Authors’ response: We thank the Reviewer for mentioning this. We have fixed the issue with the line spacing.
The table 2 must be improved.
This information of table 2 "Abbreviations: HBD, hydrogen bond donors; HBA, hydrogen bond acceptors; Log P, the logarithm of compound partition coefficient between n-octanol and water; HIA, human intestinal absorption; Pgp, P-glycoprotein; BBB, Blood-Brain Barrier; CYP3A4, Cytochrome P450 3A4" in foot of the table.
Authors’ response: We are not sure what specifically the Reviewer is asking us to do about Table 2. Table 2 gives a summary of the in silico drug-likeness, absorption, distribution, metabolism, and excretion profile of the opioid peptides from bovine and human milk. The Table footnote are intended to explain the abbreviations used in the column headings of the table.
The line 217 erase Lipinski.
Authors’ response: This has been done.
The conclusion must be improved. Erase line 481-483. Most of…
Authors’ response:
Our conclusion gives an overview of the key findings reported in the study. We are not sure exactly what the Reviewer wants us add or modify.
Lines 481 – 483 is where we mention “Figure S1: Analytical RP-HPLC chromatogram and ESI-MS of β-casomorphin 5 and β-casomorphin 7.” We feel it is important to have this in the manuscript, to give further information about the characteristics of the peptides used in the in vitro analysis of this study.

Reviewer 3 Report
Ji and colleagues present an in silico and in vitro evaluation of milk protein-derived opioid peptides for the control of Alzheimer's disease. The paper is well written and all the experiments needed to confirm the author's hypothesis are present. I will only suggest changing the peptides and the background colors in figure 3 in order to make the peptides more visible. I also suggest removing the SMILE column in table 1 to make the table more compact and easy to read.
Author Response
Reviewer 3
Ji and colleagues present an in silico and in vitro evaluation of milk protein-derived opioid peptides for the control of Alzheimer's disease. The paper is well written and all the experiments needed to confirm the author's hypothesis are present.
Authors’ response: We thank the Reviewer for the positive comments on our manuscript.
I will only suggest changing the peptides and the background colors in figure 3 in order to make the peptides more visible.
Authors’ response: We thank the Reviewer for this comment. We have regenerated the docking plots with a white background to improve visibility of the peptides.
I also suggest removing the SMILE column in table 1 to make the table more compact and easy to read.
Authors’ response: We have removed the SMILEs for the peptides, as suggested by the Reviewer.

Reviewer 4 Report
This paper focus on the effect of cholinesterase inhibition of opioid peptides derived from milk on Alzheimer disease using in computational analysis and in vitro assay. Authors show the possibility to predict potential of opioid peptides from milk to control Alzheimer disease via acethylcholinesterase inhibition. However, I don’t know how cholinesterase inhibition alleviates the Alzheimer disease. You have to explain the mechanism of cholinesterase inhibition with respect to Alzheimer more detail. A weakness of peptide for drug-likeness is a low oral bioavailability, fast degradation in vivo, fast excretion in the liver and kidney, and a low selectivity for receptor. Your peptides also have all shortcoming as peptide drug. Therefore, peptide mimetics is needed for drug-likeness. You have to try the peptide mimetics in the future.
Line 97: Did you perform the clustering of peptide? The number of clustering is decreased in long chain peptide compared to the short chain peptide. The different 3D structure of peptide results in different docking score.
Line 225: “30% HIA% value” is very confusing; how about “30% of HIA value(%)”
Line 230: Low HIA value indicates low Caco-2 cell permeability.
Line 253: Lactoferroxin B also have a positive BBB in Table 2.
Line 439: What does mean “when data from and Table 6 are compared”?
Line 447: You indicate a failure of the docking model to reflect wet experimental results. Why don’t you apply other docking methods?
The ligand peptides in Fig. 3 were expanded to out of biding site compared to those in Fig. 2. Did you consider this problem to affect the molecular docking?
Pleas, check the references according to editors’ indication.
